# A Review on Cancer Immunotherapy and Applications of Nanotechnology to Chemoimmunotherapy of Different Cancers

**DOI:** 10.3390/molecules26113382

**Published:** 2021-06-03

**Authors:** Safiye Akkın, Gamze Varan, Erem Bilensoy

**Affiliations:** 1Department of Pharmaceutical Technology, Faculty of Pharmacy, Hacettepe University, 06100 Ankara, Turkey; akkinsafiye@gmail.com; 2Department of Vaccine Technology, Hacettepe University Vaccine Institute, 06100 Ankara, Turkey; gamzevaran@hacettepe.edu.tr

**Keywords:** cancer, nanoparticle, immunotherapy, macromolecule delivery, targeting

## Abstract

Clinically, different approaches are adopted worldwide for the treatment of cancer, which still ranks second among all causes of death. Immunotherapy for cancer treatment has been the focus of attention in recent years, aiming for an eventual antitumoral effect through the immune system response to cancer cells both prophylactically and therapeutically. The application of nanoparticulate delivery systems for cancer immunotherapy, which is defined as the use of immune system features in cancer treatment, is currently the focus of research. Nanomedicines and nanoparticulate macromolecule delivery for cancer therapy is believed to facilitate selective cytotoxicity based on passive or active targeting to tumors resulting in improved therapeutic efficacy and reduced side effects. Today, with more than 55 different nanomedicines in the market, it is possible to provide more effective cancer diagnosis and treatment by using nanotechnology. Cancer immunotherapy uses the body’s immune system to respond to cancer cells; however, this may lead to increased immune response and immunogenicity. Selectivity and targeting to cancer cells and tumors may lead the way to safer immunotherapy and nanotechnology-based delivery approaches that can help achieve the desired success in cancer treatment.

## 1. Introduction

Cancer is a complex disease characterized by uncontrolled growth of cells and expansion of these abnormal cells in the body, which caused over 9.6 million deaths worldwide in 2017 alone [1]. Regarding the estimation of American Cancer Society for 2021, specialists consider that nearly 1.9 million people will be diagnosed with cancer and almost 608,570 people will die from cancer [2]. The cancer-related mortality in the world is expected to reach 22 million by the year 2030.

Among the various approaches employed in cancer treatment, surgery is usually the first treatment of choice. The appropriate strategy for treatment is developed according to the type and stage of cancer. Other treatment methods in clinical practice are chemotherapy and radiotherapy and combinations of these approaches. Chemotherapy using various antineoplastic agents can be applied as first line treatment for therapeutic purposes, but also to prevent the proliferation of cancer cells after surgery or radiotherapy or to reduce the size of tumor tissue prior to surgery. Today, although it is possible to treat cancer with one treatment method alone or with a combination of different treatment methods, the desired success rate in cancer treatment has not yet been achieved. The main reason for this failure is the systemic toxicity and undesired side effects caused by the treatment strategy, particularly for chemotherapy. In order to overcome these side effects and provide more effective cancer treatment with lower doses of active ingredient, different treatment strategies have been developed. Immunotherapy, which has attracted attention in recent years, can be defined as the use of immune system features in cancer treatment [3]. In other words, immunotherapy is defined as a type of biotherapy and is based on the sensitization of the patient’s immune system to cancer, which increases selectivity and reduces side effects [4]. The prevalence of cancer immunotherapy in clinical trials led to the selection of cancer immunotherapy as the “2013 Breakthrough of the Year” by *Science* [5].

As with other treatment methods, immunotherapy is subject to some limitations because it may impair immune homeostasis by causing an immune reaction that should not be employed against normal tissue. For instance, inflammation, colitis and pruritus can be observed frequently depending on dosage of drugs and personal factors [6]. However, positive results are obtained in immunotherapy studies that target minimum side effects compared to conventional treatments. Therefore, the advances and improvement in immunotherapy through nanotechnology and nanoparticulate delivery systems may have a positive impact on treatment success.

In this review, cancer immunotherapy is explained, and the clinical applications are detailed. In addition, the role and privileges of immunochemotherapy and nanoparticulate delivery systems in immunotherapy are summarized in the light of preclinical studies and recent literature.

## 2. The Immune System and Its Role in Cancer Immunotherapy

The immune system consists of innate and adaptive immune components (Table 1). The innate arm is the first defense mechanism against the antigen and is responsible for generating sudden and short-lived responses with monocytes, macrophages, dendritic cells and natural killer cells. Innate system cells are in charge of recognition of non-host cells and the presenting of cells of antigenic nature to adaptive system cells. Innate immune system cells have receptors to recognize microorganisms, damaged cells and transformed cells, i.e., tumor cells.

On the other hand, the adaptive arm produces long-lasting responses using T cells and B cells, eventually generating immune memory [7]. Adaptive immunity is based on T and B-lymphocytes that proliferate after recognition and destroy antigenic structures by stimulating various mechanisms. Besides these cells, the immune system’s constituents include cytokines, antibodies, plasma and adhesion molecules, tissues and organs such as the thymus, bone marrow and spleen [8].

Cancer growth and metastasis are able to exert immunosuppression mechanisms that prevent cancer from being recognized by immune system. Thus, cancer immunotherapy aims to activate the immune system against these malignant cells [5]. Immunochemotherapy is orchestrated to stimulate patient’s own immune system passively or actively to recognize and destroy damaged or abnormal cells. Cancer immunotherapy is significantly advantageous in preventing recurrence by regulating the tumor microenvironment and reducing the presence of tumor neoantigen [5]. The immune system tries to detect the cancer cells and nascent tumors and this is called “immune surveillance of cancer”. The detection and destruction of damaged or cancer cells by the immune system is a complex pathway that develops as a result of the coordination of many cells. The most important component of this multistep process is tumor-associated antigens (TAAs). TAAs are released from dead tumor cells and greatly occupy the tumor surface and are presented to T cells with antigen-presenting cells (APCs) such as dendritic cells (DC), macrophages and B lymphocytes. The main function of DCs of the innate system is to recognize microorganisms, damaged cells and transformed cells, such as cancer cells, using Toll-like receptors (TLRs) and nucleotide-binding oligomerization domain (NOD)-like receptors [9]. TLRs can be expressed on tumor cells as well as on other immune cells such as T cells and MDSCs. In particular, TLR-4 can detect damaged cell pathogen-associated molecular patterns and damage-associated molecular patterns [10]. Another component that plays a role in this field is TLR-9, which is expressed on DCs and NK cells. Clinical trials on stand-alone or combination therapy based on TLR-9 are ongoing [11]. After recognition of a foreign or harmful structure, the DCs retain this antigenic structure to present and activate the adaptive immune system cells by modifying their own surface with antigens and presenting them to the T cell receptors via the class I and II major histocompatibility complex (MHC) in the lymph nodes [12,13]. Class I MHC molecules can induce CD8⁺ T cells, which are called cytotoxic T cells (Tc) [14], that are able to destroy tumor cells and consequently offer more TAAs to enhance the immune response. This pathway is called “Cancer-Immunity Cycle” (Figure 1) [15]. Other member of the MHC molecule, class II MHC molecule, also has epitopes to be recognized by CD4⁺ T cells, which are T helpers (Th) [14]. After recognition, T cells are activated and proliferated, and the cells which have these tumor specific epitopes are called effector T cells.

The immune system attempts to eradicate cancer cells even if the tumor continues to grow and differentiate. This pathway is the process of immuno-regulation and involves different cell types and factors [15]. For example, T cells may emit inhibitory signals, such as Natural killer (NK) cells, stimulatory or regulatory T cells (Tregs) [12]. Tregs produce high levels of TGF-β and interleukin-10 (IL-10) to inhibit cytotoxic T-lymphocytes and APCs in addition reduce the immunogenic effect by reducing ATP assembly in the extracellular environment [16]. Similar to Tregs, myeloid-derived suppressor cells (MDSC) do not have a direct anti-tumor effect, but they reduce arginine amino acid, nitric oxide synthase and interleukin-2 (IL-2) levels by expressing arginine I in effector T cells [17]. Moreover, in cancer treatment, different cytokines are used for the induction of DCs and NK cells and proliferation of T cells, especially cytotoxic T lymphocytes (CTLs). Cytokines released from many different cells are the main mediator molecules in immune system cells and regulate the immune and inflammatory responses. Cytokines are a large family that consist of many proteins such as interleukins (ILs), colony-stimulating factors (CSFs), interferons (IFNs), tumor necrosis factors (TNFs) and transforming growth factor (TGF-β). In cancer treatment, different cytokines are used for the induction of DCs and NK cells and the proliferation of T cells, especially cytotoxic T lymphocytes (CTLs). In clinical use for the treatment of cancer, high-dose cytokine administration is required to induce tumor regression [18,19].

The tumor microenvironment is characterized by enhanced vascularization, hypoxia, reduced pH and increased interstitial blood pressure besides the immune cells [20]. This complex feature is a barrier restricting drug permeation into solid tumors and the activation of immune cells in tumor microenvironment. Moreover, the tumor microenvironment plays an important role in tumor development, and especially metastasis, by protecting itself from the immune system through various mechanisms. In this regard, the most widely known mechanism is that the presence of hypoxia induces tumor-associated macrophages and promotes tumor progression. (TAMs) [21]. Besides, tumor cells can protect themselves from the immune system by reducing the levels of MHC class I molecules or inducing mutation in MHC class I genes. In addition, tumors are able to express B-cell lymphoma 2 (Bcl-2) gene to inhibit apoptosis and act as signal transducer and activator of transcription 3 (STAT3) as oncogenic molecules [22].

Tumor metastasis is a multistage process in which malignant cells spread from the primary tumor to the organ via blood and lymphatic circulation. Basically, tumor growth and metastasis are caused by angiogenesis and lymphangiogenesis triggered by chemical signals from tumor cells. Therefore, cancer cells change the normal immune mechanism and try to avoid the immune system by regulating new antigens and proteins (Figure 2) [23]. For example, tumors change the DC phenotype through their microenvironment and the change in DC phenotype causes the differentiation of MDSC and M2 macrophages. As a result of this differentiation, the activation of antitumor CTL and Th1 cells is inhibited and the release of growth factors necessary for the tumor increases [24]. Another immunosuppressive agent available in the tumor microenvironment is the M2-like type macrophage stimulated for IL-10, arginase I, and TNF-secretion [15].

The strategies developed by the tumor to escape the immune system are based on blocking the activity of T cells because when the T cell recognizes the tumor, antitumor activity is stimulated. Suppressing the production of antigens necessary for the T cell to recognize that the tumor is a way of hiding the tumor from the immune system. The tumor that avoids antigen production cannot be recognized by T cells and antitumor activity cannot be triggered. The tumor can also inhibit antigen processing by causing mutations in genes such as MHC gene required for antigens [25]. In addition, the tumor microenvironment may exhibit immunosuppressive properties due to various cytokines such as vascular endothelial growth factor (VEGF), interleukins, and transforming growth factor beta (TGF-β), secreted by both cancer and immune system cells.

Tumorigenesis is a very complex process in which many metabolic changes occur, from cancer cell differentiation to metastasis. Changes in this process are in favor of tumor development. The tumor metabolism is specialized to provide the energy and factors, such as protein and DNA, necessary for the rapid division and proliferation of cancer cells and their survival. Significant changes occur in the mitochondria of cancer cells, both to prevent apoptotic effects and to provide the necessary energy in a short time. Unlike healthy cells, lactate dehydrogenase [4] accumulation is observed in cancer cells that prefer glycolysis even in the presence of oxygen. This metabolic change, known as the “Warburg Effect”, is known to be caused by oncogenes. The high concentration of LDH accumulated in the cancer cell causes direct suppression of the immune system by inhibiting CTLs and preventing the conversion of monocytes to DCs. This is another strategy for cancer cells to escape from the immune system [26].

## 3. Strategies in Cancer Immunotherapy

The efficacy of cancer immunotherapy has been demonstrated by clinical studies as well as in vitro and in vivo studies [22]. There are many different immunotherapeutic drugs approved by the Food and Drug Administration (FDA) to be used in cancer therapy: recombinant-human-IL-2 for renal cancer cell therapy; first monoclonal antibody for B-cell malignancies; first DC-based cancer vaccine for prostate cancer therapy; chimeric antigen receptor (CAR)-engineered cell therapy for B-cell lymphoma; and programmed death ligand-1 (PD-L1) immune checkpoint blockers for melanoma are examples of immunotherapeutics approved for use in the treatment of cancer in the last thirty years [27]. The evaluation of more than 40 antibodies in cancer are continued in late-stage clinical studies.

On the other hand, immunotherapeutics administered in combination therapy can be considered a reasonable and effective way to treat cancer [12]. The FDA and European Medicines Agency (EMA) approved two vaccines—T-VEC (Imylgic^®^) to induce DC production in melanoma and Sipuleucel-T (Provenge^®^) to supply genes expressing granulocyte-macrophage colony-stimulating factor (GM-CSF) in metastatic prostate cancer, respectively [28]. More than 80 antibody-based products are already approved by the FDA and EMA [29]. Furthermore, ipilimumab, pembrolizumab, nivolumab, atezolizumab, durvalumab, cemiplimab and avelumab are immune checkpoint inhibitors (ICIs) approved by the FDA and authorized by the EMA in the last seven years [30]. For unresectable or metastatic hepatocellular carcinoma, atezolizumab and bevacizumab are approved as combined therapy by the FDA in 2020 [31]. Meanwhile, the FDA has promoted different combinations of immunotherapy, such as ipilimumab plus T-VEC, pembrolizumab plus T-VEC, ipilimumab plus peptide vaccine, nivolumab plus peptide vaccine and pembrolizumab plus viral vaccine in melanoma, and ipilimumab plus Sipuleucel-T in prostate cancer [32].

Cancer immunotherapy strategies are basically divided into four types (Figure 3): immunomodulation, adoptive cellular therapy, targeted antibodies and cancer vaccines.

(A)**Immunomodulation** focuses on improving the immune response by priming the host immune system. Although different agents play a role in immunomodulation, the basic approach is based on the stimulation of antigen presenting cells to T cells and consequently the further killing of tumor cells by T cells. Cytokines are the leading immunomodulation agents. There are several cytokine-based immunomodulators approved for use in immunotherapy of different cancer types. For example, Aldesleukin (Proleukin^®^) is a synthetic form of IL-2 produced by recombinant DNA technology and is approved for the treatment of kidney cancer and melanoma [33]. Another example is interferon alfa-2b (Intron A^®^), also produced by recombinant technology and used in the treatment of leukemia, lymphoma and melanoma [34].

Another immunomodulator group used in cancer immunotherapy are checkpoint inhibitors. Checkpoint inhibitors act by blocking the proteins that prevent the immune system from attacking cancer cells. According to their mechanism of action, they can also be defined as targeted therapy, because checkpoint inhibitors inhibit specific proteins on the T cell or cancer cell. For example, Ipilimumab (Yervoy^®^), developed to inhibit CTLA-4 (cytotoxic T lymphocyte associated protein 4) expressed on T cells, is used in the treatment of advanced melanoma. Normally, CTLA-4 is a protein receptor that acts as an immune checkpoint that downregulates immune responses. In this way, it prevents the immune system from attacking healthy cells in autoimmune diseases such as rheumatoid arthritis and ulcerative colitis. However, in the treatment of melanoma, the aim is to increase the attack of T cells to the tumor by inhibiting this CTLA-4 receptor [35]. PD-L1 is one of the cell surface proteins expressed on cancer cells that functions as an immune system suppressor. Atezolizumab (Tecentriq^®^), developed as a PD-L1 inhibitor, is used in the treatment of lung cancer and urothelial cancer [36]. Although a limited patient group benefits from immune checkpoint blockers with close to 40% response rates in the clinics, the effect depends on the T cell levels at the tumor site. For this reason, these agents are combined with other chemotherapeutics, radiotherapeutics or vaccines. In addition, recent studies have shown that some chemotherapeutic agents, such as Doxorubicin, have induced host immune system when used alone or combined with immunotherapy. Moreover, the research on the efficacy of doxorubicin and immune checkpoint blockers is ongoing in clinical trials (NCT04028063). It is known that doxorubicin can induce antitumor T cell responses that induce immunogenic cell death by releasing calreticulin (CRT) and high mobility group box 1 (HMGB1) [27,37]. In a study, αPD-L1 and αCTLA-4 antibodies combined with IL-2 in collagen-binding domain (CBD). Tumor volume was reduced in the melanoma, colon, and breast tumor models in the CBD-IL2-treated group. However, when CBD-ICI and CBD-IL2 were administered together, it was emphasized that the tumor was almost completely destroyed in the breast tumor model and a survival rate of 69% was achieved after 100 days [38]. In another study CBD combined with the chemokine CCL4 and its effectiveness was evaluated in multiple tumor models. In addition to increased tumor accumulation, overexpress of DC and T cells was been reported [39]. IL-12 was combined with CDB to minimize its adverse effects and its efficacy was evaluated after intravenous administration in a melanoma model. As a result of the study, it was reported that IFN-y, ALT and AST levels were decreased compared to free IL-12. While short-term tumor shrinkage was observed when IL-12 was administered alone, a long-term antitumoral response of nearly 60% was obtained when IL-12and ICI were administered in combination [40]. In addition, clinical trials are ongoing on new immune checkpoint inhibitors. For example, success in the treatment of solid tumors is observed in phase I studies with antibodies targeted to CD47, CD73, A2Ar and TIM-3 [41].

(B)**Adoptive cell therapy** is a type of immunotherapy applied to help the immune system fight cancer cells. In cellular immunotherapy, T cells are used in different ways by taking advantage of their natural abilities to eliminate cancer cells (Figure 4). Basically, T cells are collected from the cancer patient’s blood or tumor tissue and then changed in the laboratory to better target the cancer cells and then given to the patient [42]. The presence of T cells may not always be sufficient to eliminate cancer cells. Killer T cells must also be present in a sufficient number of tumor sites, be pre-activated and maintain their activity until the tumor is completely eliminated. Tumor-Infiltrating Lymphocyte (TIL) therapy, one of the adoptive cell therapy applications, is an immonutherapy approach that aims to provides all of these conditions. In TIL treatment, T cells that have already infiltrated the tumor tissue of the patient with cancer are collected, expanded and re-infused into the patient in order to provide a sufficient number. Despite the promising benefits of TIL therapy, it has some limitations. Unfortunately, although T cells are reproduced in vitro conditions, sometimes sufficient numbers are still not achieved in patients. To overcome this problems, engineered T cell receptor (TCR) therapy using peripheral lymphocytes has been developed with an genetic engineering approach. This approach not only activates and expands existing anti-tumor T cells, but also enables the T cells to target specific cancer antigens. In both TIL and TCR treatment approaches, only cancer cells presenting their antigens can be targeted. T cells can recognize cancer thanks to the MHC-mediated presentation of specific antigens expressed on the surface of cancer cells. In the chimeric antigen receptor (CAR-T) approach developed to overcome this limitation, T cells recognize cancer cells as MHC-independent. This approach is an example of personalized medicine practice and the FDA and EMA have also approved CAR-T therapeutic products Kymriah^®^ and Yescarta^®^ for use in lymphoma treatment [43].

(C)**Targeted Antibodies:** Monoclonal antibodies (mAbs) can be briefly defined as antibodies that bind to specific parts of an antigen. The high specificity of monoclonal antibodies binding to cancer cells has made it inevitable that it would be used in cancer treatment. mAbs have been researched for use in cancer treatment for many years and there are a variety of products on the market. The idea of benefiting from active targeting in cancer treatment has made the use of mAbs inevitable. In particular, the discovery of tumor-specific antigens has increased the interest in mAbs. The main reason for using mAbs in chemotherapy is that systemic toxicity can be prevented. For this reason, versatile research continues for the use of mAbs in cancer treatment. Rituximab, developed for CD20 expressed on the surface of B cells in non-Hodgkin lymphoma, is the first mAb used in cancer treatment. Subsequently, anti-HER2 Trastuzumab for breast cancer treatment and anti-VEGF Bevacizumab and anti-EGFR Cetuximab for colorectal cancer treatment were approved. To date, approximately 30 mAbs are FDA-approved for use in cancer treatment and are on the market [44,45]. mAbs can be used alone in cancer treatment or in combination with anticancer agents. This technological approach, called antibody–drug conjugate, aims to carry anticancer drugs to the tumor via mAbs produced specifically on the cancer cell surface antigen. In this way, a form of chemotherapy with increased efficiency and reduced side effects is provided [46]. Kadcyla is an antibody–drug conjugate approved by the FDA and EMA in 2013 for use in the treatment of HER2-positive metastatic breast cancer [47]. It is a combination of anti-HER2 Trastuzumab and DM1, a chemotherapy drug effective on microtubules. Adcetris is a conjugation of Brentuximab (chimeric IgG1) that targets CD30 on the cancer cell surface and the cytotoxic agent monomethyl auristatin E. It was approved by the FDA in 2011 and then by the EMA in 2012 for use in the treatment of Hodgkin lymphoma [48]. The mAbs and antibody–drug conjugate that have been approved for use in cancer treatment are summarized in Table 2.

The previously mentioned approaches (adoptive cellular therapies, targeted antibodies and immunomodulators) used in cancer immunotherapy are grouped as passive immunotherapy. In passive immunotherapy, ex vivo activated cells are used to stimulate the immune system against cancer cells.

(D)**Vaccine** use in cancer immunotherapy is an example of active immunotherapy. In active immunotherapy, the aim is to activate the effector functions of the immune system [49]. The main purpose of cancer vaccines containing tumor cells or antigens is to stimulate the immune system, destroy the tumor and prevent relapse. Cancer vaccines are responsible for introducing specific antigens expressed on the surface of cancer cells to the immune system. With the elucidation of the structures of cancer cell specific tumor-associated antigens, interest in cancer vaccines is increasing day by day. These antigens are combined with adjuvants and used to activate the immune system against cancer. Cancer vaccines are developed to prevent or treat cancer. For example, there are vaccines for human papillomavirus (HPV), which is associated with different types of cancer such as cervical, vaginal and throat is done to prevent cancer. Similarly, the Hepatitis B (HBV) virus can be used to reduce the risk of hepatocellular carcinoma in patients with chronic liver disease. In cancer vaccines developed for therapeutic purposes, adjuvants that help increase the immune response are generally used. Sipuleucel-T (Provenge), the first commercially available cancer vaccine, is a dendritic cell-based vaccine developed for the treatment of hormone-refractory prostate cancer [50]. Talimogene laherparepvec (T-VEC) is a vaccine used in the treatment of melanoma and is an oncolytic herpes simplex virus developed with a genetic engineering approach. It is also the first FDA-approved oncolytic viral therapy product [51,52].

## 4. Nano-Immunotherapy Applications in Cancer Treatment

Nanomedicines benefit from the present innovative and targeted approaches to the selective treatment of cancer. With this technology, several nano-sized drug carrier systems have been developed to overcome biodistribution and pharmacokinetic problems of active pharmaceutical ingredients (APIs). The use of nanoparticulate systems in cancer started in 1995 with the FDA approval of Doxil^®^ (PEGylated liposomal doxorubicin). Then, in 1996, Doxil^®^ was approved by the EMA under the trade name Caelyx [53]. Since then, FDA and EMA approved nanosystems have increased rapidly. About 55 nanomedicines are currently used in clinical settings and more than half of them have reached the market for cancer therapy. The combination of two innovative strategies in cancer treatment, immunotherapy and nanotherapy, are promising fields. Currently, there are over 100 clinical studies using combinations of different immunotherapeutic and nanocarriers. Besides the clinical studies, the combination of atezolizumab (Tecentriq) with Abraxane was approved by the FDA for use in the treatment of metastatic triple negative breast cancer [54]. However, atezolizumab is not approved for use with Taxol (paclitaxel). This strategy is the first approved example of the combined application of nanomaterials with biological therapeutics in cancer treatment.

Nanoparticulate carrier systems have been investigated for use in cancer treatment for many years with several advantages as a result of their unique structures (Figure 5) listed as follows:Nanoparticles interact directly with the cell membrane and intracellular structures, avoiding the drug resistance mechanism by bypassing the cellular efflux pump and increasing uptake into tumor cells.Nanoparticulate systems increase the cellular uptake and accumulation of the transported drugs into the tumors.Nanosystems enhance the efficiency and stability of active molecules by protecting them from biological factors such as enzymes during blood circulation.The surface of nanosize carriers can be modified to active targeting or also targeted passively thanks to their particle sizes. Long-term circulation of these nanoparticles can strengthen the targeting ability.Nanomedicines provide enhanced immune responses in comparison to the use of conventional immunotherapy alone.

Nanoparticulate systems increase the response of the immune system through various mechanisms, i.e., by targeting specific immune cells, activating and enabling immune cells to recognize cancer cells [53]. By using nanocarrier systems, it is possible to protect the active molecule from biological factors and to deliver the drugs effectively to cancer tissues. Thanks to these advantages, nanoparticles are used in cancer nano-immunotherapy for the delivery of cytokines, vaccines, immune checkpoint inhibitors, agonistic antibodies and engineered T cells [13]. Another advantage of using nanoparticles in cancer immunotherapy is that systemic toxicity can be eliminated with these delivery systems. Immunotherapy may cause severe side effects and toxicity commonly characterized by skin reactions such as pain, swelling, redness, itchiness, rash and soreness. In addition, gastrointestinal side effects such as diarrhea, vomiting, colitis and also neurotoxicity, endocrine problems and pneumonia are the other adverse effects of conventional immunotherapy [14,55]. Nanosystems can help in reducing the amount of immunotherapeutic required for treatment. Moreover, they can help the active ingredient to accumulate at the tumor site avoiding other organs to overcome systemic side effects [21,56]. Besides active targeting, it is also possible to benefit passive targeting in cancer immunotherapy. As in chemotherapy, the enhanced permeability and retention effect due to the natural structure of the tumor microenvironment also enables the use of nanoparticles in immunotherapy by using passive targeting. Different nanoparticles, such as polymeric, lipid-based or metallic particles, are used as delivery systems for genes and drugs in cancer immunotherapy. Examples of studies performed in this field in the last 5 years are summarized in Table 3 and Table 4.

The purpose of this review is to summarize the applications of cancer immunotherapy with a focus on nanosystems. In this context, the most recent nano-immunotherapy approaches for the most common types of cancer have been compiled.

### 4.1. Nano-Immunotherapy of Melanoma

Melanoma has the fastest increase in incidence among cancer types. Current treatment includes surgery, chemotherapy, targeted therapy or radiation therapy [76]. Immunotherapy is generally used in the treatment for stage III and IV melanoma. Treatment with immunotherapy may use immune checkpoint inhibitors (Keytruda^®^ and Opdivo^®^), cytokines (IFN and IL-2), or imiquimod cream. Cancer vaccines, which are another immunotherapy approach used in the treatment of melanoma, attract great attention. There are many vaccines for this purpose in clinical trials. One of these vaccines is NeoVax (NCT01970358), a personalized neoantigen cancer vaccine consisting of peptide plus poly-ICLC (Polyinosinic-Polycytidylic acid stabilized with polylysine and carboxymethylcellulose, also known as Hiltonol^®^). Another example of a vaccine, NEO-PV-01 (NCT02897765), consists of nivolumab and an adjuvant and is evaluated for the treatment of melanoma, bladder and lung cancers [77]. IVAC^®^ Mutanome (NCT02035956) has been investigated for stage III and IV melanoma patients. It is the first fully personalized RNA vaccine concept. Reportedly, vaccination with IVAC^®^ Mutanome was very well tolerated and showed high immune-enhancing effect [78]. Clinical studies have shown that immunotherapy can be a good treatment option especially in the spread and invasion of melanoma cells in lymph nodes. However, the expected therapeutic response from conventional immunotherapy fails due to high cytotoxicity to immune cells such as monocytes, macrophages, NK cells, hepatocytes and neurons [55]. In such cases, biochemotherapy and new delivery systems, such as those that combine nanoparticles and immunotherapy or chemotherapy, gain importance [79].

As mentioned above, nanoscale systems are utilized to improve the effectiveness of treatment by eliminating toxicity caused by conventional immunotherapy [80,81,82]. For example, the transport of IL-2 with adenovirus was developed by modifying T cells as stimulants and inducers of NK and B cells. In order to overcome the immunogenicity caused by the adenovirus, a new carrier system needs to be developed. For this purpose, Yao et al. prepared β-cyclodextrin and folate conjugate polyethyleneimine polyplexes as non-immunogenic vehicle for IL-2. This nanosystem was designed as a polycationic vector. For this purpose, cationic charge was achieved by using polyethyleneimine. In addition, active targeting is achieved through modification with folic acid. It was emphasized that the toxicity caused by multiple dosing, which is required in the IL-2 treatment regimen, can also be avoided thanks to active targeting. The in vivo studies revealed that dose-dependent antitumoral effect was observed in melanoma-induced mice. The activities of the viral vector and polyplex as IL-2 carriers were reported to be equivalent and that this system could be considered an alternative strategy for melanoma treatment [83]. In a study about combination immunotherapy approaches for melanoma treatment, Park et al. designed a liposomal gel system to co-deliver TGF-β receptor I inhibitor (SB505124) and IL-2 to overcome the immunoinhibitory nature of the tumor microenvironment. TGF- β receptor I inhibitor was incorporated into the nanoparticulate system by forming an inclusion complex with methacrylate-β-cyclodextrin to solubilize the TGF-β inhibitor. Additionally, IL-2 encapsulating a polylactic acid-polyethylene glycol-polylactic acid (PLA-PEG-PLA) nanoparticle combined with the drug: the cyclodextrin inclusion complex in the liposomal gel system has a particle size of nearly 120 nm. It was reported that the nanosystem delayed tumor growth, improved survival rate and therapeutic benefits by inducing NK cells and CD8⁺ T cells in the tumor microenvironment. It was also shown that sustained and slow release of IL-2 is possible, and that a sufficient dose of IL-2 can be provided for the treatment of metastatic melanoma [84].

Some anticancer drugs have immunomodulatory effects and increase efficacy in melanoma treatment when used together with immunotherapeutics [85]. In a study performed to benefit from the immunogenic efficiency of Paclitaxel and IL-2-loaded biomimetic chitosan nanogel formulation was prepared and modified with erythrocyte membrane coating material to achieve nanosponge properties. According to the results, it was reported that chitosan nanogel provides increased antitumoral effects by enhancing the inductive effect of dendritic cells, cytotoxic T lymphocytes and immunostimulatory cytokine secretion [79]. In another study, paclitaxel and TLR-4 agonist were encapsulated into poly (lactic-co-glycolic acid) (PLGA) nanoparticles showing a prolonged release profile. The nanoparticles had an equivalent cytotoxic effect on melanoma cell lines with lower concentrations as regards the paclitaxel solution. Furthermore, melanoma-induced mice studies showed that the APCs and T cells were activated, and tumor volume was reduced with higher secretion of IL-2, IFN-γ, TNF-α and IL-1β compared to drug solution [86].

### 4.2. Nano-Immunotherapy of Colorectal Cancer

As estimated by the World Health Organization (WHO) for 2018, colorectal cancer (CRC) was the third most common cancer based on new cases and the second most common cancer-related cause of death in world. Although surgery is first choice treatment, chemotherapy is also applied before and after surgery according to the stage of cancer in the patient. In addition, targeted therapy and radiation therapy is applied in clinical settings [87]. In immunotherapy, another approach in the treatment of CRC see the use of targeted antibodies such as Avastin^®^ and Cyramza^®,^ and ICI applications such as Keytruda^®^ and Opvido^®^ are among the treatment regimens in the clinic [88].

Considering the approaches to nanoparticulate delivery systems, not only mono-immunotherapy but also combined therapy are investigated, as shown in Table 3 and Table 4. Besides these examples, Bousheri et al. developed lipopolysaccharide (LPS) loaded PLGA nanoparticles and investigated the effects of LPS on TLR-4 structure, which is a component of innate immunity and induces cytokine synthesis. The efficiency of LPS loaded nanoparticles and LPS solution was compared in the murine colorectal cancer model. According to the findings, it was observed that nanoparticles decreased the systemic toxicity of LPS and increased TNF-α and IL-6 levels compared to the LPS solution. On the other hand, it was observed that PLGA nanoparticles prevent the recurrence of tumors when injected into peritumors in mice [21]. As mentioned before, some chemotherapeutic agents have an immunogenic effect in addition to the antitumor response. For instance, the anticancer drug Oxaliplatin, used in the treatment of colorectal cancer, may enhance the immune response by increasing CD8⁺ T cells expression by lowering Myeloid-derived suppressor cells and Treg levels. It is also known that when these agents are combined with immune adjuvants such as antibodies or cytokines, immune activation is induced in the tumor [89]. In the literature, self-assembled core-shell nanosystems loaded with Oxaliplatin and dihydroartemisinin were combined with anti-PD-L1 antibody therapy. This nanosystem containing oxaliplatin loaded ZN nanoparticles is covered with a lipid bilayer containing cholesterol-DHA conjugate. It was observed that prolonged antitumor immunity was achieved by inducing T cell activation in murine colorectal tumors and by reducing suppressor cell infiltration [90].

Exosome is membranous vesicles and function as cell–cell communication mediator. In biology and pharmacology, these systems are used as therapeutic carrier systems for RNA and drugs to treat several diseases. In this context, Lugini et al. developed exosomes from human colorectal cancer inducing mesenchymal stromal cells that play promoter roles of tumor progression. Therefore, it was suggested that tumor growth inhibition might be carried out by stopping exosome formation in the tumor [91]. On the other hand, exosomes are used as vehicles for the delivery of therapeutics to treat colorectal cancer. Additionally, vaccine studies containing exosomes which have been derived from dendritic or tumor cells have been performed in treatment of colorectal cancer and many cancers [92]. In addition, researchers presented exosome-mimetic nanovesicles derived from M1 macrophages (M1NVs). Once M1 types macrophages are activated, tumor growth is suppressed and tumor vascularization is normalized. Actually, TAMs in tumor microenvironment, is alternatively activated M2 types, which foster tumor growth and angiogenesis by releasing anti-inflammatory cytokines. M1NVS are formed to repolarize M2 macrophages to M1 types and investigated whether the two systems augment ICI therapies with anti-PD-L1 or not. In vivo studies were carried out in CT26-bearing mice showed that M1NVs have no toxic effects on major organs and they accumulate at the tumor area. Treatment with M1NVs and a-PD-L1 decreased tumor volume by inducing a form of apoptotic cells compared to M1NVs or a-PD-L1 alone. Moreover, q-PCR studies showed that the combination upregulated pro-inflammatory genes and M1 markers, such as CD86 and CD80, also reduced anti-inflammatory genes and M2 markers [93].

### 4.3. Nano-Immunotherapy of Lung Cancer

Constituting almost all of the lung cancer cases, non-small cell lung cancer (NSCLC) is commonly treated with surgery. However, sometimes chemotherapy and radiotherapy are used alone or combination as well. In the early stages of this cancer, different radiation treatments can be preferred, such as brachytherapy or radiofrequency ablation. Advanced stage lung cancer is treated with chemotherapeutic agents that are targeted to gene mutations such as ALK, BRAF and receptors such as EGFR and VEGFR [94]. CIMAvax- EGF^®^ was approved by the Cuban Regulatory Authority in 2008 and is known as the first vaccine for the treatment of stage IIIB and IV lung cancer. CIMAvax-EGF^®^ has been approved in recent years and used in Argentina, Bosnia and Herzegovina, Colombia, Kazakhstan, Paraguay and Peru. The vaccine contains human recombinant EGF, a recombinant carrier protein and an adjuvant Montanide ISA51. CIMAvax-EGF^®^ aims to prevent interaction with EGFR by producing antibodies against EGF [95]. On the other hand, a clinical trial investigated the effectiveness of CIMAvax- EGF^®^, nivolumab and pembrolizumab on NSCLC as well as head and neck squamous cell cancer (NCT02955290). In this phase I/II trial, it was reported that administration of drugs as cancer vaccine was well tolerated by patients and provided more effective treatment [96].

Another vaccine candidate is AST-VAC2, an allogeneic cancer vaccine targeting the regulation regulate telomerase level of cells. It is developed by Cancer Research UK in phase I/IIa for NSCLC treatment and possible side effects and safety problems are investigated (NCT03371485). Another allogeneic cellular vaccine is 1650-G, which is combined with a beta glucan oral supplement in phase I/II clinical trials (NCT01829373). It was reported that a safe and strong immunogenic response was observed in this study. On the other hand, the TUSC1 nanoparticle gene delivery system contains the FUS1 gene that induces apoptosis and is absent or reduced in most of the lung cancers. The nanoparticle is being investigated for the effectiveness of its combination with Tarceva^®^ in NSCLC patients in phase I/II clinical trials (NCT01455389).

The main reason for using nanoparticles in the treatment of lung cancer is to provide a reliable and effective treatment regimen by eliminating off-target cytotoxicity, as in other cancer types [97]. Zhang et al. evaluated the effectiveness of glycocalyx-mimicking nanoparticles self-assembled into amphiphilic copolymers to target tumor-associated macrophages (TAMs) in Lewis lung cancer induced mice. In the study, TAMs were collected from the tumor tissue of mice and it was observed that nanoparticles were internalized through the lectin receptor to reduce activity and were affected by the reversal of TAMs. These glycocalyx-mimicking nanoparticles were reported to enhance the specific interaction with lectin receptors due to high concentrations of carbohydrates. When tumor-induced mice were injected intravenously with α-PD-L1 and subcutaneous nanoparticles, an increase in the expression of IL-12 and a decrease in the level of IL-10, arginase I (a cross-reactivity mechanism is of genetically coded for protected immunity), and CCL22 (macrophage-derived chemokine responsible of activating T cells). Nanoparticles also inhibited tumor growth by depleting TREGs [22].

### 4.4. Nano-Immunotherapy of Lymphoma

Lymphomas are commonly classified into two types as Hodgkin lymphoma and Non-Hodgkin lymphoma (NHL). However, lymphoma has many subtypes depending on the type of cell from which the cancer originated, such as B-cells or T-cells. Both these types of lymphomas are treated with alkylating agents, antibiotics, corticosteroids and/or radiotherapeutic agents [98]. In addition, immunotherapy is used as a new strategy in the clinic, such as CAR-T cell therapy, monoclonal antibodies, immunomodulatory agents and ICI. In lymphoma, tumor cells can spread from the lymph node to the extranodal regions. Hence, conventional chemotherapy is often insufficient to reach tumor cells. To overcome this problem, new studies have been performed for immunotherapy alone or combined with chemotherapy by using nanotechnology. Vaccine studies are also frequently investigated in this area [99,100]. Notably, there are several studies with CpG, which is used as vaccine adjuvant to target lymph nodes in the literature. In a study, it was aimed to combine a dendritic cell (DC) targeting vaccine and chemotherapy to be used in lymphoma treatment. The adjuvant was formed by combining CpG with the cationic polymer chitosan. CpG is a TLR-9 agonist and enhances cytokine synthesis after inducing DCs. After transfection in tumor cells, the adjuvant is formulated with mannose to target DC cells. It is known that mannose is phagocytosed via receptors on the DC surface. It was determined that vaccine formulation has an improved targeting effect. The effectiveness of the system was demonstrated in the lymphoma treatment model as well as in the lymphoma prevention model. For this purpose, the vaccine was combined with the anticancer drug doxorubicin. The combined system showed stronger antitumoral efficacy compared to chemotherapy and immunotherapy [100]. In another study, the efficiency of PEG-Fmoc-NLG (PEG-derivatized NLG919) micelles loaded with doxorubicin in lymphoma was investigated. NLG (NLG919) is an indoleamine 2,3-dioxygenase 1 (IDO-1) inhibitor agent. It was reported that nanoparticles increase CD4⁺/CD8⁺ T cells and IFN-γ levels as a result of the intravenous administration in A20 lymphoma mouse model. In addition, IC_50_ values of nanoparticle formulation, free doxorubicin and Doxil^®^ were compared and it was seen that nanoparticles had the lowest value with 0.58 μg/mL. In contrast to free doxorubicin solution, cardiotoxicity was not observed in the nanoparticle treated groups [101].

### 4.5. Nano-Immunotherapy of Breast Cancer

The treatment of breast cancer is mainly based on chemotherapy and breast conserving surgery (lumpectomy) or mastectomy. Radiotherapy is an alternative strategy used instead of breast conserving surgery. Studies are also ongoing for hormone therapy in the treatment of breast cancer as well as immunotherapy and targeted therapy [102]. NeuVax^TM^ is a cancer vaccine that contains HER-2 derived peptides and targets HER-2 positive breast cancer cells. This vaccine was combined with Leukine^®^ (GM-CSF: Granulocyte-macrophage colony-stimulating factor) in a Phase III clinical trial in 2016 and compared to GM-CSF (NCT01479244) for disease-free survival. Additionally, in three different clinical studies, the NeuVax^TM^ and trastuzumab and/or GM-CSF vaccine combinations were in phase II, NCT02297698, NCT02636582, NCT01570036, respectively. In these studies with trastuzumab, it was seen that there was a significant clinical benefit in treatment. In another study, AVX901 containing trastuzumab to target HER-2 was shown to be safe in phase I (NCT01526473) and continued phase II clinical trial (NCT03632941) with pembrolizumab.

The existence of nanomedicines such as Abraxane^®^ and Myocet^®^, which are approved for use in the treatment of breast cancer, makes the application of nanoparticulate systems indispensable in developing alternative strategies in breast cancer treatment. In this regard, both natural and synthetic nanoparticles are frequently investigated for use in nano-immunotherapy [103,104]. To overcome the challenges of antibody therapies such as safety, efficacy and bioavailability, Colzani et al. incorporated trastuzumab into PLGA biodegradable nanoparticles. Thus, it was aimed to protect trastuzumab against pH conditions. With SDS-PAGE, circular dichroism and fluorescence emission methods, it was shown that trastuzumab is released from nanoparticles while preserving structural integrity. The released trastuzumab was evaluated for selectivity for binding to HER-2 in SKBR3 cell lines by flow cytometry and confocal microscopy. In addition, trastuzumab and doxorubicin loaded PLGA nanoparticles were prepared to reduce the risk of tumor recurrence and this immunochemotherapeutic system showed higher cytotoxicity on tumor cells compared to the groups treated with free doxorubicin solution and doxorubicin loaded PLGA nanoparticles [105].

As mentioned above, another example of the use of exosomes as biocompatible therapeutic cargo is focused on breast cancer. Exosomes were used in this study aiming to kill cancer cells by activating cytotoxic T cells. It has been reported that exosomes expressing monoclonal antibodies specific for T cell CD3 and cancer cell-associated epidermal growth factor receptor (EGFR) elicit strong antitumor immunity. In vivo studies showed a specific and strong toxicity on MDA-MB-468 triple negative breast cancer cells via interaction with T cells and EGFR. Moreover, it was observed that tumor growth was reduced in exosome treated animals when compared with PBS-treated animals [106].

### 4.6. Nano-Immunotherapy of Hepatocellular Carcinoma

Liver cancer is one of the common cancer types worldwide mostly caused by hepatitis B virus (HBV) and/or hepatitis C virus (HCV), excessive alcohol consumption, diabetes, obesity and smoking. The HBV vaccine has been used to treat liver infection since 1982 but there is no vaccine for hepatocellular carcinoma yet. The treatment strategies for hepatocellular carcinoma are surgery, liver transplantation, embolization, radiotherapy, chemotherapy and targeted therapy drugs such as Nexavar^®^, Lenvima^®^ and Stivarga^®^ [107]. Additionally, hepatic artery invasion, which is a direct application of chemotherapeutic agents into hepatic artery to enhance efficiency of drugs, is applied clinically. The immunotherapeutic drugs Opvido^®^ and Keytruda^®^ are used in unresectable and metastatic liver cancer to help control the tumor growth [108,109].

A vaccine clinical trial has been performed with HepaVac-101, which consists of RNA-based immunomodulating agent in phase I/II. This vaccine was combined with cyclophosphamide and was evaluated for early as well as intermediate stage of hepatocellular carcinoma (NCT03203005). Targeted therapy and combined therapy with immunotherapy or nanotechnology have been studied to improve survival rates of liver cancer patients. For instance, doxorubicin and recombinant human IL-2 was combined into chitosan nanocomplexes having a particle size of around 200 nm and surface modification with folate. According to in vitro findings, it was found that activity and structure of rhIL-2 was protected in nanocomplexes. Moreover, nanocomplexes enhanced cellular uptake of doxorubicin and decreased cell viability on SMMC-7711 cell lines in comparison to corresponding free drugs and polymer solutions. The in vivo studies showed that the mice had unchanged body weight, reduced hemolysis ratio in liver and tumor volume, and increased IgG level and tumor infiltrated lymphocytes in the nanocomplex treated group [110].

### 4.7. Nano-Immunotherapy of Gynecological Cancers

Ovarian cancer is the fifth leading cause of cancer-related death among women globally. Surgery and chemotherapy are generally used in clinical settings for treatment of ovarian cancer. On the other hand, immunotherapy (bevacizumab) and hormone therapy (anastrozole, letrozole or tamoxifen) are preferred in stage IV and at recurrence. Cervical cancer is the second leading cause of cancer death in women aged 20 to 39 years [111]. Surgery, radiation, chemotherapy, targeted therapy such as bevacizumab (Avastin^®^) and Keytruda^®^ may implement as treatment regimen in ovarian and cervical cancer. There are three HPV vaccines—Gardasil^®^, Gardasil 9^®^ and Cervarix^®^— which are approved by the regulatory authorities and used in clinical settings [112]. In addition, Gardasil 9^®^ was approved for the prevention of oropharyngeal and head and neck cancer for males and females between the ages of 9 and 45. A new vaccine is Depovax^TM^, which completed phase I study as DPX-Survivac. The system is a liposome-based DC vaccine combined with low dose cyclophosphamide (oral) for ovarian cancer, fallopian tube and peritoneal cancer [113].

Ovarian cancer is considered a “cold” tumor because of its limited response to immunotherapies. The most basic feature of tumors categorized as “cold” is the absence of T cells in the tumor microenvironment [114]. This leads to the need to develop systems that can provide more effector cells to eradicate the tumor. Nanoparticulate systems are used to overcome the deficiencies in this field and to provide more effective immunotherapy [115]. In a study aimed to block the programmed death ligand-1 (PD-L1), which is highly expressed in ovarian cancer cells and causes immunosuppression. Accordingly, PD-L1 siRNA loaded folic acid (FA) modified polyethyleneimine nanoparticles were prepared. Modification with FA was reported to increase cellular uptake of nanoparticles, as these cells also overexpress folate receptors. Moreover, it has been stated that as a result of the inhibition of PD-L1, T cells become more sensitive to tumors [74].

### 4.8. Nano-Immunotherapy of Bladder Cancer

Bladder cancer is estimated to rank fourth among new cases of cancer in men [111]. Surgery is the main option in the treatment of bladder cancer. Immunotherapy, chemotherapy, and radiotherapy are the other treatment regimens used in clinical settings [116]. Additionally, ICI such as nivolumab, pembrolimumab, and avelumab are approved by the FDA for bladder cancer. There are also combination studies in which immunotherapeutics or chemotherapeutics have been evaluated in many different clinical trials [117].

BCG (Bacille Calmette-Guérin) immunotherapy is reported as the standard treatment for bladder cancer [118]. In the case of nanoparticulate immunotherapeutic systems, BCG was loaded into the liposome using emulsified liquid evaporation. Although BCG has a very high molecular weight and negative charge, the liposomal system has been successfully developed with a particle size of 280 nm. In vivo studies in tumor-induced rats have shown that antitumor efficacy is higher than free BCG in terms of tumor weight reduction [119]. In another study, Erdoğar et al. formulated BCG loaded chitosan nanoparticles with a particle size of 375 nm using an ionotropic gelation technique. In this study, the aim was to develop nanoparticles suitable for intravesical drug delivery, which is a mucosal application. In intravesical administration, the thick mucus gel layer (Bladder Permeability Barrier) covering the bladder mucosa is a barrier that must be overcome. For this purpose, chitosan nanoparticles with cationic surface charge and penetration enhancers were used in the study. In vivo studies in tumor-associated rats have shown that nanoparticles had higher antitumor activity than free BCG solution without any adverse effects on the lung tissues [120].

In addition, the BCG cell wall skeleton (BCG-CWS) is used as an adjuvant in nanoparticulate systems. Masuda et al. showed that BCG-CWS nanoparticles (CWS-NP) induced antitumoral and immune effects after intravenous administration in mice. In that study, it was determined that CWS-NP application promoted dendritic cell maturation and cytokine stimulation. Moreover, it was observed that CWS-NP increased the antitumoral efficacy of OVA-NP in combination therapy [121]. In another study, BCG-CWS nanoparticles were encapsulated in liposomes in order to reduce the side effects of BCG treatment. As a result of this study, increased antitumor activity was also shown with the induction of autophagy and ROS production [122].

## 5. Conclusions

Although there are various alternative treatment regimens, cancer still remains as the second leading cause of death globally according to a 2021 WHO report. Many approaches are used in clinical settings to eliminate cancer, such as chemotherapy, radiotherapy and surgery. Immunotherapy is more recent than these therapies and includes cellular therapy, antibody therapy, vaccine therapy, and nonspecific therapy. Nanotechnology-based therapies have been launched as novel and more selective therapeutics for cancer treatment. For an eventual improved efficacy, nanoparticulate drug delivery system combined immunotherapeutics are being developed both in vitro and clinically. As a result of advances in cell biology and tumor biochemistry, the structures of cancers and tumors are increasingly being clarified. In particular, as the tumor microenvironment and immunosuppressive mechanisms are enlightened, the success of immunotherapy increases. The idea of destroying the tumor by re-stimulating the immune system and its elements, which naturally exist but are inhibited in tumor development, gains importance day by day. It is possible to treat cancer by using effective and less active molecules with nano-immunotherapy. Moreover, nano-immunotherapy studies emphasize that personalized medicine is possible, important and inevitable in cancer treatment. As in other fields, the use of nanoparticulate systems in cancer immunotherapy is expected to provide selectivity and enhancement of antitumoral effects and will be the cancer treatment strategy of the future.

## Figures and Tables

**Figure 1 molecules-26-03382-f001:**
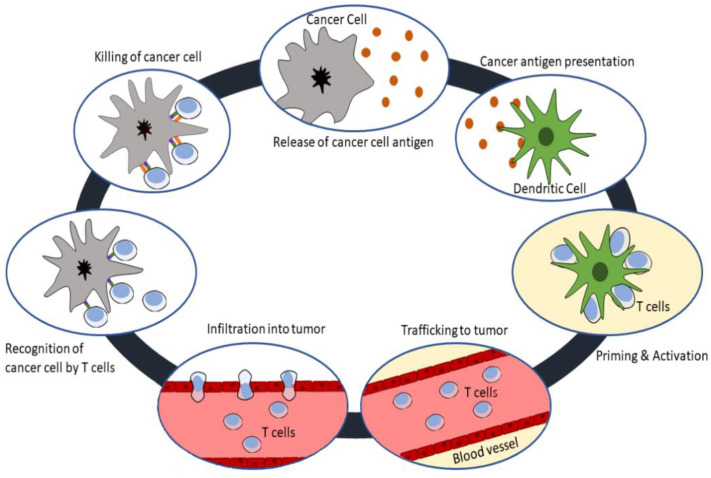
The cancer immunity cycle. This cycle starts with the release of antigens from cancer cells and ends with the formation of immunity to the cancer cell, and many different cells play a role in this natural process.

**Figure 2 molecules-26-03382-f002:**
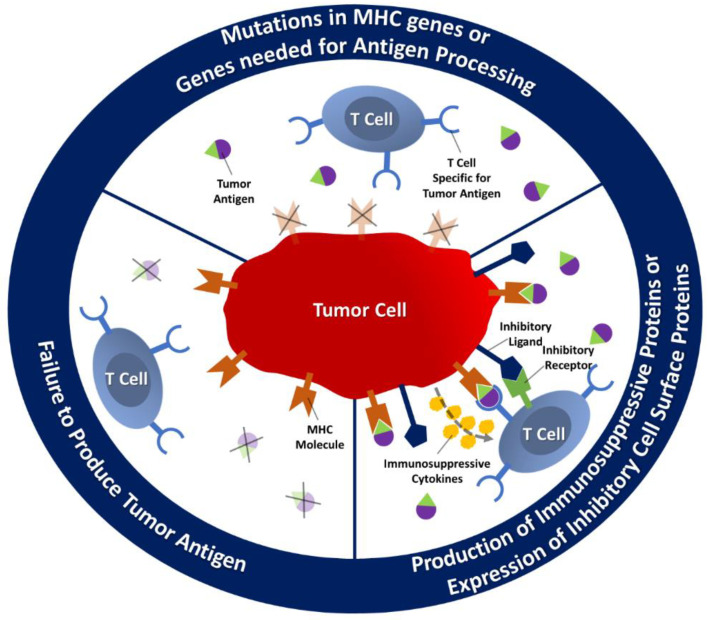
Mechanism of tumor escape from the immune system.

**Figure 3 molecules-26-03382-f003:**
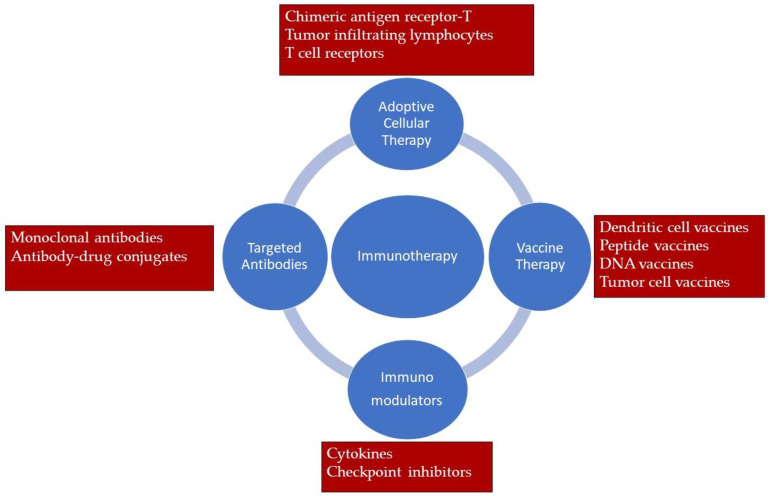
Immunotherapy approaches for cancer treatment.

**Figure 4 molecules-26-03382-f004:**
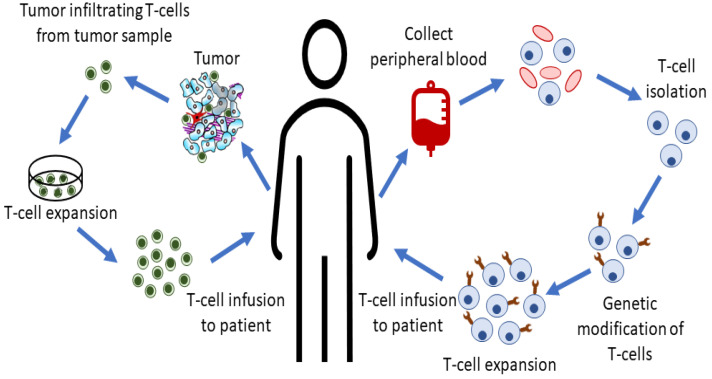
Adoptive cell therapy approaches.

**Figure 5 molecules-26-03382-f005:**
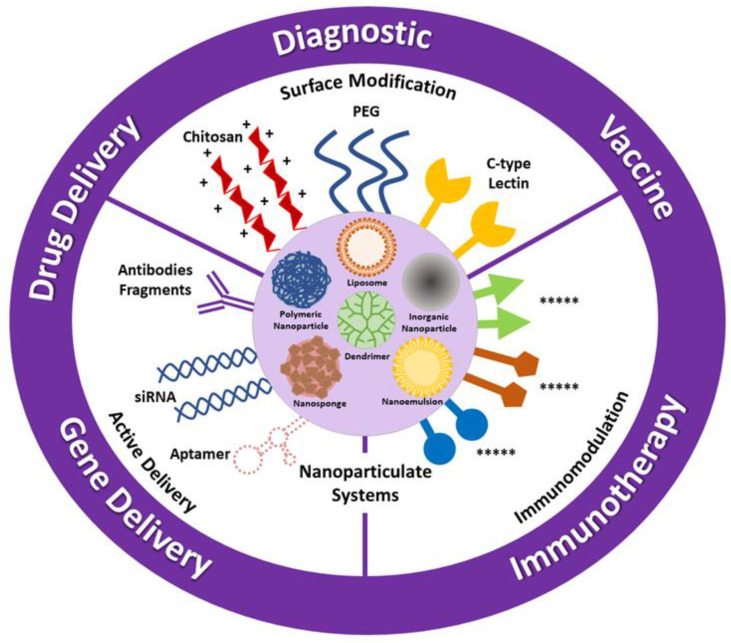
Various types of nanoparticulate delivery systems and possible approaches in cancer therapy.

**Table 1 molecules-26-03382-t001:** Immune system types and constituents.

Innate Immunity	Adoptive Immunity
Epithelial barriers	B Lymphocytes
Phagocytes (Macrophage ex.)	T Lymphocytes
Mast cells	
Dendritic cells	
Natural killer and other innate lymphoid cells	
Complement proteins	

**Table 2 molecules-26-03382-t002:** Summary of monoclonal antibodies used in the treatment of cancer in the clinic.

Name	Target	Antibody/Cytotoxic Agent	Indication(s)
Rituximab (MabThera^®^, Rituxan^®^)	CD20	Human–mouse chimeric IgG1κ	Non-Hodgkin’s lymphoma
Ibritumomab (Zevalin^®^)	CD20	Murine IgG1κ linked chelator tiuxetan	Non-Hodgkin’s lymphoma
Ofatumumab (Arzerra^®^)	CD20	Fully human IgG1κ	B-cell CLL
Obinutuzumab (Gazyva^®^, Gazyvaro^®^)	CD20	humanized mAb of IgG1κ	CLL
Isatuximab (Sarclisa^®^)	CD38	Chimeric IgG1	Multiple myeloma
Daratumumab (Darzalex Faspro^TM^)	CD38	Human IgG1	Multiple myeloma
Necitumumab (Portrazza^®^)	EGFR	Human IgG1	Non-small cell lung cancer
Cetuximab	EGFR	Chimeric IgG1	Colorectal cancer; Head and neck squamous cell carcinoma
Panitumumab	EGFR	Human IgG2	Colorectal cancer
Bevasizumab	VEGF	Humanized IgG1	Colorectal cancer; Non-small cell lung cancer; Renal cancer; Glioblastoma; Ovarian cancer
Ramucirumab	VEGFR2	Human IgG1	Gastric cancer
Pertuzumab	HER2	Humanized IgG1	Breast cancer
Trastuzumab	HER2	Humanized IgG1	Breast cancer

**Table 3 molecules-26-03382-t003:** Summary of several nanoparticulate drug delivery systems on cancer immunotherapy.

Nanocarrier	Chemotherapeutic Load	Immunotherapeutic Load/Cell	Therapeutic Indication	Findings
Thermosensitive PLGA nanoparticle	Doxorubicin	IFN-γ	Melanoma	Increased cytokine levels and drug circulation [57]
Immunoliposome	-	Fab segment of anti-PD-L1	Melanoma	Similar antitumoral effect with lower dose than anti-PD-L1 and Long circulation time [12]
Poly (propyl acrylic acid) nanoplex	-	Decalceine modified antigenic peptide	Melanoma	Increased activation of CD8⁺ T cells and prolong antigen uptake [58]
ChitosanNanoparticle	-	OVA	Melanoma	Promoted DC maturation, induced antigen specific CD8⁺ T cells and increased anticancer efficacy [59]
PEG-b-PAEMA pH-sensitive cluster nanoparticles	Platinum prodrug	BLZ-945	Colon cancer	Reduced tumor volume and TAM level [60]
High density lipoprotein nanodisc	Doxorubicin	Anti PD-1	Colon cancer	Higher antitumor effect No toxicity and liver metastasis [27]
Lipid nanocapsule (MPB-DOPE and DOPG)	SN-38	Functionalized T cells with SN-38 nanocapsule	Lymphoma	Highly concentrated in lymph node than free drug, reduced tumor burden and enhanced survival rate [61]
Polyglycerol and Cyclic tripeptides of L-arginine, glycine and L-aspartic acid nanodiamond	Doxorubicin	Nano-DOX delivery was provided with DC- mediated.	Glioblastoma	Activation of DC and lymphocytes, Stimulated glioblastoma cells’ immunogenicity [62]

Abb: MPB-DOPE (1,2-dioleoyl-sn-glycero-3- phosphoethanolamine-N-[4-(p-maleimidophenyl) butyramide]; DOPG (1,2- dioleoyl-sn-glycero-3-phospho-(1’-rac-glycerol); SPC (Soy phosphatidylcholine); (DSPE-PEG-Mal) 1,2-Distearoyl-sn-glycero-3-phosphoethanolamine-N[maleimide(polyethyleneglycol)-2000] (ammonium salt); OVA ovalbumin.

**Table 4 molecules-26-03382-t004:** Summary of several nanoparticulate gene delivery systems on cancer immunotherapy.

Nanocarrier	Gene System	Combined Drug	Therapeutic Indication	Findings
Multi-walled carbon nanotubes	OVA AntiCD40 CpG	-	Melanoma	Enhanced antigen specific T cells, reduced tumor volume [63]
Polyethyleneimine nanoplexes	VEGFR-2 encoded DNA	-	Melanoma	Increased IL-2, IFN-γ, TNF-α Reduced tumor volume, enhanced survival rate [64]
Lipid nanoparticle (DOPE and Cholesterol)	p-DNA	Anti PD-L1	Melanoma	Enhanced antitumoral effect [65]
Lipoplex (DOTAP and m-PEG-PLA)	p-IL-15	-	Colon cancer	Increased IL-15 and TNF-α, induced apoptosis in tumor and inhibited tumor cell proliferation [66]
1,2-dioleoyl-3-trimethylammonium propane and dioleolylphosphatidyl ethanolamine nanocomplex	SGT-53 plasmid	Anti PD-L1	Lymphoma	Enhanced immune response, reduced tumor volume and lung metastasis [67]
Poly (propylene sulfide) and Dextran nanoparticle	CpG	OVA	Lymphoma	Stronger CD8⁺ T cells activation [68]
DOTAP lipid nanoparticle	p-CXCL12 and p-PD-L1	-	Pancreatic cancer	Enhanced accumulation in tumor, increased CD8⁺ T cells, IL-12a, TNF-α, IFN-γ level [17]
Cationic polylactic nanoplexes	IL-8 siRNA	-	Prostate cancer	Increased IL-8 expression, Reduced tumor volume [69]
Multi-walled carbon nanotubes	CpG	-	Prostate cancer	Increased T cells and IL-6 level and suppressed tumor growth [70]
PEGylated lipid polyplex	p-IL-15	-	Hepatocellular carcinoma	Increased lymphocytes, IFN-γ, IL-12 [71]
Lipid nanoparticle	IL-12 mRNA	-	Hepatocellular carcinoma	Augmented CD4⁺ T cells Reduced tumor burden [72]
Gal-C-dextran nanocomplex	CpG AntiIL-10 AntiIL-10R	-	Hepatocellular carcinoma	Induced TAMs mediated immune activity Enhanced tumor accumulation [73]
Folic acid- functionalized PEI nanoparticle	PD-L1 siRNA	-	Ovarian cancer	Knockdown of PD-L1 Increased sensitivity cancer cells [74]
Nanodisc (ApoA-I peptide, Sphingomyelin and Cholesterol)	CpG	Doxorubicin	Glioblastoma	Enhanced tumor regression and survival rate [75]

## Data Availability

Not applicable.

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
