# Peer review of "A Review on Cancer Immunotherapy and Applications of Nanotechnology to Chemoimmunotherapy of Different Cancers"

_molecules, 2021, doi:10.3390/molecules26113382_

Round 1

Reviewer 1 Report

The authors summarizes the current development of nano-immunotherapy, and provides valuable information for other researchers, such as the basics of cancer immunotherapy and nanosystems in immunotherapy.  The content of this review is well organized and comprehensive. 

A few suggestions:

  1. Comment of the clinical status of nanosystem for immunotherapy will be helpful.
  2. In Part 4 Nano-Immunotherapy Applications in Cancer Treatment, from line 389 to 410, this paragraph need be reorganized. 

Author Response

Dear Ms. Kayla Li,

We would like to thank the referees for their valuable comments and contribution to our manuscript. The manuscript was revised accordingly and responses to the referees specific comments are as follows:

Referee 1

1.Comment of the clinical status of nanosystem for immunotherapy will be helpful.

We would like to thank the referee for this comment. Atezolizumab+Abraxane, the first approved example of the combined application of biological therapeutics and nanomaterials in cancer therapy, was added in the text. In each section, the clinical studies related to that subject were already included in the text beforehand.

2.In Part 4 Nano-Immunotherapy Applications in Cancer Treatment, from line 389 to 410, this paragraph need be reorganized. 

This formatting error has been revised in the text.

Reviewer 2 Report

This manuscript describes the recent developments in cancer immunotherapy, especially by using nanoparticulate systems. The author summarized the basics in cancer immunity, categories of cancer immunotherapy and nanoparticles-based cancer immunotherapeutic applications in various cancer models. Current manuscript well covers from basic immunology to cancer treatments via modulating of immune systems. However, current manuscript still contains some points to be addressed. The list of comments is as follows:

-Title should be changed because almost half part of this manuscript is spared to basic immunology and non-nanoparticulate delivery system-based cancer immunotherapy.

-Line93-94: Is it correct that “DCs recognized damaged cells and transformed cells using TLRs? Please specify which TLRs recognize damaged cells and transformed cells with literatures.

-Carefully check that figure numbers are correct.

-In “immunomodulation” section, recent achievements in engineered cytokines should be cited such as Sci Transl Med. 2019 Apr 10;11(487):eaau3259., Sci Adv. 2019 Dec 11;5(12):eaay1357., Nat Biomed Eng. 2020 May;4(5):531-543. Furthermore, another checkpoint inhibitor in clinical trials such as CD47 antibody should be described.

-Quality of each figure should be improved.

-Sections 4.1-4.8, the author should explain the molecular design of each nanoparticulate system to improve the therapeutic effect specifically for each cancer model.

-In section 4.8, another BCG-based nanoparticulate system such as Mol Pharm 15: 5762-5771 (2018) should be cited.

Author Response

Dear Ms. Kayla Li,

We would like to thank the referees for their valuable comments and contribution to our manuscript. The manuscript was revised accordingly and responses to the referees specific comments are as follows:

Referee 2

1.Title should be changed because almost half part of this manuscript is spared to basic immunology and non-nanoparticulate delivery system-based cancer immunotherapy.

The authors thank the referee for this valuable comment which will help convey the content of the manuscript better in the title. According to reviewer coment, the title of review was revised as “A review on cancer immunotherapy and  applications of nanotechnology to chemoimmunotherapy of different cancers”

2.Line93-94: Is it correct that “DCs recognized damaged cells and transformed cells using TLRs? Please specify which TLRs recognize damaged cells and transformed cells with literatures.

As per reviewer comment and to better explain the TLRs, this section was revised as follows and added to the manuscript.

“TLRs can be expressed on tumor cells as well as on other immune cells such as T cells and MDSCs. In particular, TLR-4 can detect damaged cell pathogen-associated molecular patterns and damage-associated molecular patterns [10]. Another that plays a role in this field is TLR-9, which is expressed on DCs and NK cells. Clinical trials on stand-alone or combination therapy based on TLR-9 are ongoing [11].”

3.Carefully check that figure numbers are correct.

The figure numbers were checked.

4.In “immunomodulation” section, recent achievements in engineered cytokines should be cited such as Sci Transl Med. 2019 Apr 10;11(487):eaau3259., Sci Adv. 2019 Dec 11;5(12):eaay1357., Nat Biomed Eng. 2020 May;4(5):531-543. Furthermore, another checkpoint inhibitor in clinical trials such as CD47 antibody should be described.

Thanks to the referee for mentioning these publications to be referrred. These studies performed by Ishihara et al. were added in the “immunomodulation” section. In addition, clinical trials were also added.

5.Quality of each figure should be improved.

The quality of each figure was adjusted to the resolution recommended by the journal.

6.Sections 4.1-4.8, the author should explain the molecular design of each nanoparticulate system to improve the therapeutic effect specifically for each cancer model.

Authors would like tothank the referee for this valuable comment. To better explain the potential of each nanoparticle type in immunotherapy, structure or molecular design of the correspomding nanoparticulate systems were explaned in the text.

7.In section 4.8, another BCG-based nanoparticulate system such as Mol Pharm 15: 5762-5771 (2018) should be cited.

The authors thank the referee for this comment. Another examples, including the suggested publication, were added to the relevant section as follows:

“In addition, BCG cell wall skeleton (BCG-CWS) is used as an adjuvant in nanoparticulate systems. Masuda et al. showed that BCG-CWS nanoparticles (CWS-NP) induced antitumoral and immune effects after intravenous administration in mice. In thatstudy, it was determined that CWS-NP application promoted dendritic cell maturation and cytokine stimulation. Moreover, it was observed that CWS-NP increased the antitumoral efficacy of OVA-NP in combination therapy [121]. In another study, BCG-CWS nanoparticles were encapsulated in liposomes in order to reduce the side effects of BCG treatment. As a result of this study, increased antitumor activity was also shown with the induction of autophagy and ROS production [122].”

Round 2

Reviewer 2 Report

The author revised the manuscript almost properly according to the reviewer’s comments. Current manuscript is acceptable for publication.